# Inhibition of Lipopolysaccharide-Induced Inflammatory Signaling by Soft Coral-Derived Prostaglandin A_2_ in RAW264.7 Cells

**DOI:** 10.3390/md20050316

**Published:** 2022-05-09

**Authors:** Osamu Ohno, Eika Mizuno, Junichiro Miyamoto, Tomoyuki Hoshina, Takuya Sano, Kenji Matsuno

**Affiliations:** Department of Chemistry and Life Science, School of Advanced Engineering, Kogakuin University, 2665-1 Nakano, Tokyo 192-0015, Japan

**Keywords:** LPS, soft coral, PGA_2_, RAW264.7

## Abstract

Lipopolysaccharide (LPS) is a component of the outer membrane of Gram-negative bacteria and causes inflammatory diseases. We searched MeOH extracts of collected marine organisms for inhibitors of LPS-induced nitric oxide (NO) production in RAW264.7 cells and identified prostaglandin A_2_ (PGA_2_) as an active compound from the MeOH extract of the soft coral *Lobophytum* sp. PGA_2_ inhibited the production of NO and reduced the expression of inducible NO synthase (iNOS) in LPS-stimulated RAW264.7 cells. Although short preincubation with PGA_2_ did not inhibit LPS-induced degradation and resynthesis of IκBα, the suppressive effect of PGA_2_ was observed only after a prolonged incubation period prior to LPS treatment. In addition, PGA_2_-inhibited NO production was negated by the addition of the EP4 antagonist L161982. Thus, PGA_2_ was identified as an inhibitor of LPS-induced inflammatory signaling in RAW264.7 cells.

## 1. Introduction

Lipopolysaccharide (LPS), one of the most common and potent pathogenic factors in human blood, is an endotoxin derived from the outer membrane of Gram-negative bacteria [1]. When it is released from bacterial cell walls into the blood, LPS binds to Toll-like receptor (TLR) 4, which is expressed in innate immune cells, including macrophages, neutrophils, and natural killer (NK) cells. TLR4 is a pattern-recognition receptor that recognizes the molecular patterns associated with pathogenic compounds such as LPS. LPS activates TLR4 signal transduction, including the nuclear factor-κB (NF-κB) and mitogen-activated protein kinase (MAPK) pathways [2,3,4,5]. In particular, NF-κB plays an important role in the development of inflammatory responses by the production of proinflammatory mediators, nitric oxide (NO), and cytokines in macrophages. Activated NF-κB induces the upregulation of inducible NO synthase (iNOS) and the production of NO from the amino acid L-arginine. NO is a signaling molecule that plays a key role in the pathogenesis of inflammation [6]. Furthermore, TLR4 signaling is activated in various inflammatory diseases induced by LPS [7,8]. Thus, searching for molecules that inhibit LPS-induced NO production is a promising strategy for the discovery of new anti-inflammatory agents, and several small-molecule compounds that regulate this signaling have been investigated [9,10]. Marine organisms produce a variety of structurally unique compounds and therefore are attractive sources of drug candidates. Recently, marine natural products chrysamide B and biseokeaniamide A were reported for their anti-inflammatory activity in the inhibition of LPS-induced NO production [11,12]. In this study, we searched for metabolites of marine organisms that inhibit NO production by the LPS-stimulated murine macrophage-like cell line RAW264.7. We identified (15*S*)-prostaglandin A_2_ (PGA_2_, Figure 1) from the MeOH extract of the soft coral *Lobophytum* sp. and found that it inhibits LPS-induced inflammatory signaling in RAW264.7 cells.

## 2. Results

### 2.1. Isolation of PGA_2_ from the Soft Coral Lobophytum *sp.*

We screened for inhibitors of LPS-induced NO production by RAW264.7 cells from several hundred samples of marine organisms and found that the MeOH extract of the soft coral *Lobophytum* sp., collected at the coast of Ishigaki City, Okinawa Prefecture, Japan, showed marked inhibitory activity. The active component was isolated by chromatographic separation. Spectroscopic analyses (Appendix A) revealed that the purified compound was identical to PGA_2_ (Figure 1) [13,14,15]. PGA_2_ was recently independently isolated from the same genus of soft coral and shown to inhibit LPS-induced production of NO in RAW264.7 cells [16]. However, the mechanism of its inhibitory activity against NO production was not revealed. In the present study, we employed commercially available PGA_2_ ((15*S*)-prostaglandin A_2_; Cayman Chemical Company) and analyzed its mechanism of action.

### 2.2. Inhibition of NO Production by PGA_2_ in LPS-Stimulated RAW264.7 Cells

NO production was determined by measuring the nitrite content released into the culture media using Griess reagent. As shown in Figure 2, NO production by RAW264.7 cells could be detected after exposure to 1 µg/mL LPS for 24 h compared with the vehicle control. The addition of PGA_2_ prior to LPS stimulation significantly decreased the production of NO by RAW264.7 cells in a concentration-dependent manner (Figure 2A). The IC_50_ value was 3.19 µM. The results of an MTT assay showed no significant change in cell number after exposure to PGA_2_ in the presence of LPS for 24 h. Polymyxin B (PMB), used as a positive control, also showed a significant inhibitory effect on NO production in LPS-stimulated RAW264.7 cells (Figure 2B).

### 2.3. Inhibition of LPS-Induced iNOS Expression in RAW264.7 Cells by PGA_2_

To further evaluate the mechanisms by which PGA_2_ inhibits NO production, we examined the protein expression of iNOS by Western blotting. As shown in Figure 3, the protein levels of iNOS were significantly upregulated in response to 1 μg/mL LPS. PGA_2_ reduced the levels of iNOS in LPS-stimulated RAW264.7 cells in a concentration-dependent manner. PMB inhibited the LPS-induced expression of iNOS at a concentration of 1 μM.

### 2.4. Effects of PGA_2_ on the LPS-Induced Expression of IL-6 in RAW264.7 Cells

Proinflammatory cytokines play an important role in the activation of the inflammatory response. To evaluate the anti-inflammatory effect of PGA_2_, the contents of proinflammatory cytokine interleukin-6 (IL-6) in the culture medium of LPS-stimulated RAW264.7 cells were measured by ELISA. As shown in Figure 4, the level of IL-6 was significantly elevated by 1 μg/mL LPS. PGA_2_ showed a concentration-dependent inhibitory effect on the LPS-induced production of IL-6 in RAW264.7 cells (IC_50_ value 9.00 μM).

### 2.5. Effects of PGA_2_ on the Expression Levels of IκB in RAW264.7 Cells

LPS induces the proteasome-mediated degradation of the inhibitor of NF-κB (IκB) to activate NF-κB, which regulates the genes encoding iNOS. Activated NF-κB also binds to the IκB promoter to resynthesize IκB [17]. Accordingly, we examined the effects of PGA_2_ against the degradation and resynthesis of IκB using Western blotting. LPS induced the degradation of IκBα at 30 min (Figure 5A), then IκBα synthesis was reactivated after 90 min (Figure 5C). Pretreatment with PGA_2_ for 20 min did not inhibit the LPS-induced degradation of IκBα, as shown in Figure 5A, and 30 μM PGA_2_ did not inhibit the LPS-induced degradation of IκBα (Figure 5B). The resynthesis of IκBα was not inhibited by the presence of 10 μM PGA_2_ (Appendix A) or 30 μM PGA_2_ (Figure 5C). On the other hand, PMB clearly inhibited the LPS-induced degradation of IκBα (Appendix A). These results demonstrate that PGA_2_ does not inhibit the LPS-induced activation of NF-κB directly.

### 2.6. Inhibition of LPS-Induced Degradation of IκBα by Prolonged Treatment with PGA_2_

LPS-induced degradation of IκBα was not inhibited in RAW264.7 cells treated with PGA_2_ for 20 min. To determine whether prolonged treatment of cells with PGA_2_ affects NF-κB activation, we studied IκBα degradation in cells pretreated with 30 μM PGA_2_ for 18 h, followed by an LPS challenge. As shown in Figure 6A, PGA_2_ inhibited LPS-induced IκBα degradation at concentrations above 10 μM following treatment for 18 h. Although treatment with PGA_2_ for 18 h slightly reduced the protein levels of IκBα, LPS-induced IκBα degradation was substantially reduced in RAW264.7 cells by prolonged treatment with 30 μM PGA_2_ (Figure 6B). In addition, the expression of IκBα was unaffected by LPS-stimulation for 90 min in cells pretreated with PGA_2_ for 18 h. Thus, PGA_2_ retains the ability to suppress IκBα degradation in LPS-activated RAW264.7 cells following prolonged incubation. These results suggest that indirect mechanisms mediate the anti-inflammatory effects of PGA_2_.

### 2.7. Effects of L161982 on PGA_2_-Inhibited NO Production in LPS-Stimulated RAW264.7 Cells

A previous study identified the PGE_2_ receptor EP4 as a receptor of PGA_2_ in human pulmonary endothelial cells [18]. Therefore, in this study, we investigated the effect of an EP4 antagonist, L161982, on PGA_2_-inhibited NO production in LPS-stimulated RAW264.7 cells. As shown in Figure 7, PGA_2_-inhibited NO production was negated by the addition of 10 μM L161982 to LPS-stimulated RAW264.7 cells pretreated with 1 μM or 3 µM PGA_2_. These results showed that PGA_2_ might inhibit LPS-induced NO production mediated by its receptor EP4 and the de novo synthesis of related proteins in RAW264.7 cells.

## 3. Discussion

In this study, we searched through the metabolites of marine organisms to identify compounds that can inhibit LPS-induced NO production by RAW264.7 cells. Screening several hundred MeOH extracts of marine organisms led us to identify PGA_2_ from the MeOH extract of the soft coral *Lobophytum* sp. PGA_2_ decreased LPS-induced NO production without cytotoxicity up to 30 µM (Figure 2A) and suppressed the expression of iNOS in LPS-stimulated RAW264.7 cells (Figure 3). Furthermore, PGA_2_ inhibited the LPS-induced production of IL-6 in RAW264.7 cells (Figure 4). PGA_2_ did not inhibit LPS-induced IκBα degradation when added 20 min prior to an LPS challenge (Figure 5); rather, the suppressive effect of PGA_2_ was observed only after a prolonged incubation period (18 h) prior to LPS treatment (Figure 6). PGA_2_-inhibited NO production was negated by the addition of the EP4 antagonist L161982 to LPS-stimulated RAW264.7 cells (Figure 7).

Prostaglandins (PGs) are lipid mediators belonging to the eicosanoid family and are key players in a wide variety of physiological and pathological processes in mammals. PGA_2_ has been reported to be produced via the metabolic dehydration of PGE_2_ in cultured mammalian cells [19]. Although PGA_2_ may be produced non-enzymatically as a rearrangement of PGE_2_, it is not clear whether PGA_2_ is present in mammals. Our current results might shed light on this matter. There are many reports on the biological activities of PGA_2_ in mammalian cells, such as the cell cycle arrest of NIH3T3 cells at the G_1_ and G_2_/M phase by PGA_2_ [20]. PGs have also been discovered in marine invertebrates [21]. The Caribbean gorgonian *Plexaura homomalla* was reported to contain 1 million times higher levels of PGA₂ than that found in most other organisms, suggesting that PGA_2_ functions in *P. homomalla* as a chemical defense against predators [22].

PGs function as intracellular signal mediators in the regulation of inflammation and immune responses. Cyclopentenone PGs, including PGA_2_, were reported to display particularly anti-inflammatory activities and to interfere with the signaling pathway that leads to the activation of transcription factor NF-κB [21]. PGA_2_ reportedly inhibits the production of NO, cytokines, and chemokines by LPS-stimulated microglia and astrocytes [23]. In addition, PGA_2_ suppresses LPS-induced inflammatory signaling by inhibiting the NF-κB pathway in human pulmonary endothelial cells. This effect was demonstrated to be mediated by EP4, a PGE_2_ receptor [18]. Indeed, PGE_2_ and an agonist of EP4 were demonstrated to inhibit the proinflammatory actions of LPS in mouse adult ventricular fibroblasts [24]. EP4 is present in RAW264.7 cells [25], but the inhibitory activity of PGA_2_ against LPS-induced inflammatory signaling in macrophages has not been investigated in detail. PGA_1_ was reported to be a potent inhibitor of NF-κB activation in human cells and acts by inhibiting the phosphorylation and preventing the degradation of IκBα. The inhibition of NF-κB does not require protein synthesis but rather is dependent on the presence of a reactive cyclopentenonic moiety [26]. The inhibitory activity of 15-deoxy-Δ^12,14^-PGJ_2_ (15d-PGJ_2_) on LPS signaling has been well studied. 15d-PGJ_2_ exhibits a potent anti-inflammatory effect by binding to p50 of NF-κB [27,28]. In addition, 15d-PGJ_2_ is a high-affinity ligand for peroxisome-proliferator-activated receptor γ (PPARγ) and has been demonstrated to inhibit the induction of inflammatory response genes, including iNOS and TNFα, in a PPARγ-dependent manner. Furthermore, the expression of the cytoprotective enzyme heme oxygenase-1 (HO-1) is induced and coincident with the anti-inflammatory action of 15d-PGJ_2_, suggesting that the expression of HO-1 contributes to the suppression of LPS-induced IκB degradation induced by 15d-PGJ_2_ [29].

The results of the present and previous studies on PGs suggest that PGA_2_ does not suppress LPS-induced NO production by direct interaction with LPS signal transduction factors. Rather, PGA_2_ might inhibit LPS-induced IκB degradation mediated by its receptor and the de novo synthesis of related proteins in RAW264.7 cells. Indeed, PGA_2_-inhibited NO production was negated by the addition of the EP4 antagonist L161982 in LPS-stimulated RAW264.7 cells (Figure 7). On the other hand, there is also the possibility that PGA_2_ may be transformed into an active form, such as PGA_1_, in order to show activity. Regardless, PGA_2_ was identified as an inhibitor of LPS-induced inflammatory signaling in the murine macrophage-like cell line RAW264.7. Further research may reveal the detailed mechanism of PGA_2_ as an anti-inflammatory agent.

## 4. Materials and Methods

### 4.1. Materials

Chemicals and solvents were of the highest grade available and used as received from commercial sources. LPS (*Escherichia coli*. sc-3535) was purchased from Santa Cruz Biotechnology, Inc. (Dallas, TX, USA). PGA_2_ ((15*S*)-prostaglandin A_2_) and L161982 were purchased from Cayman Chemical Company (Ann Arbor, MI, USA). Polymyxin B (PMB) was procured from Wako (Tokyo, Japan). Monoclonal antibody against iNOS was purchased from Abcam (Cambridge, MA, USA). Monoclonal antibodies against IκBα and α-tubulin were procured from Cell Signaling Technology (Danvers, MA, USA). NMR spectra were recorded with a JEOL HNM-ECX400 FT NMR spectrometer (JEOL, Tokyo, Japan). Mass spectra (EIMS) were obtained on a JEOL JMSGC MATE II (JEOL). Optical rotation was measured on a JASCO P-2200 polarimeter (JASCO, Tokyo, Japan) using a microcell (light path, 100 mm). HPLC was carried out using an LC-10AT_VP_ pump (Shimadzu, Tokyo, Japan) with an SPD-10AV_VP_ UV detector (Shimadzu).

### 4.2. Biological Material

Specimens of the soft coral *Lobophytum* sp. were collected along the coast of Ishigaki Island (24°27′27.3″ N, 124°09′54.0″ E), Okinawa Prefecture, Japan, in May 2018, and were frozen immediately after collection. Specimens of the soft coral were preserved at the School of Advanced Engineering, Kogakuin University.

### 4.3. Extraction and Isolation of (15S)-Prostaglandin A_2_

The soft coral *Lobophytum* sp. (420 g, wet weight) was extracted with MeOH for 2 weeks. The extract was filtered, concentrated in vacuo, and partitioned between EtOAc and H_2_O. The EtOAc layer was concentrated and further partitioned between *n*-hexane and 90% aqueous MeOH. The 90% aqueous MeOH layer, which showed remarkable inhibitory activity against LPS-induced NO production in RAW264.7 cells, was fractionated using ODS silica gel column chromatography (MeOH-H_2_O). The 80% aqueous MeOH eluent was then subjected to reversed-phase HPLC (STR ODS-II, Shinwa Chemical Industries, Ltd., Kyoto, Japan; solvent MeOH-H_2_O) to give (15*S*)-prostaglandin A_2_ (1.9 mg) as a colorless oil.

### 4.4. Cell Culture

RAW264.7 cells were cultured at 37 °C with 5% CO_2_ in DMEM (Nissui, Tokyo, Japan) supplemented with 10% heat-inactivated fetal bovine serum (HyClone; GE Healthcare Hyclone Laboratories, Logan, UT, USA), antibiotic-antimycotic mixed stock solution (1% *v*/*v*, 100 units/mL penicillin, 100 μg/mL streptomycin, 0.25 μg/mL amphotericin B; Nacalai Tesque, Inc., Kyoto, Japan), 2 mM L-glutamine, and 2.25 mg/mL NaHCO_3_.

### 4.5. Nitric Oxide Determination

NO production was measured as previously reported [30]. RAW264.7 cells were seeded at 1 × 10^6^ cells/mL in 96-well plates and cultured overnight. Then, various concentrations of PGA_2_ were added. After treatment with 1 μg/mL LPS, the cells were incubated for 24 h. An aliquot of culture medium (100 μL) was mixed with an equal volume of Griess reagent (1% sulfanilamide and 0.1% *N*-naphthylethylenediamine hydrochloride in 2.5% phosphoric acid). Optical density at 550 nm was measured with a microplate reader (Synergy H1, BioTek, Winooski, VT, USA).

### 4.6. MTT Assay

The MTT (3-(4,5-dimethylthiazol-2-yl)-2,5-diphenyltetrazolium-bromide) assay was performed on cells remaining on the 96-well plates used for NO determination. Briefly, MTT reagent (15 µL at 1.44 mg/mL; Nacalai Tesque, Inc., Kyoto, Japan) was added, and the samples were incubated for 4 h. The formazan crystals formed were dissolved in 100% DMSO, and optical density at 540 nm was measured with a Synergy H1 microplate reader.

### 4.7. Western Blotting Analysis

RAW264.7 cells (1 × 10^6^ cells) were treated with PGA_2_ and stimulated with or without LPS for the desired periods. Then, the cells were scraped off and suspended in RIPA buffer (50 mM Tris-HCl (pH 7.4), 150 mM NaCl, 2 mM EDTA, 100 mM NaF, 2 mM Na_3_VO_4_, 1% Triton X-100, 1% sodium deoxycholate, 0.1% sodium *n*-dodecyl sulfate (SDS), 1 μg/mL aprotinin, and 1 mM PMSF). The supernatants were combined with 2× sample buffer (125 mM Tris–HCl, 20% glycerol, 0.01% bromophenol blue, and 4% SDS) and 2.5% 2-mercaptoethanol, then electrophoresed in 10% polyacrylamide gels. The gels were electrophoretically transferred to polyvinylidene difluoride (PVDF) membranes (Bio-Rad Laboratories, Inc., Hercules, CA) for 30 min. The membranes were then blocked with 10% BSA and incubated with iNOS, IκBα, or α-tubulin antibody in TBS buffer (20 mM Tris-HCl (pH 7.4), pH 7.6, and 500 mM NaCl) at room temperature for 1 h. The blotted membranes were washed 6 times with 0.1% Tween 20 in TBS buffer and incubated with horseradish-peroxidase (HRP)-conjugated goat anti-rabbit IgG (Abcam) for 1 h. Immunoreactive proteins were visualized by using a luminol-based chemiluminescence assay kit (Chemi-Lumi One L; Nacalai Tesque, Inc., Kyoto, Japan) and an ImageQuant LAS 4000 (GE Healthcare Bio-Sciences AB, Uppsala, Sweden).

### 4.8. IL-6 Production Assay

RAW264.7 cells (1 × 10^5^ cells) seeded in 24-well plates were treated with PGA_2_ for 20 min and stimulated with or without 1 μg/mL LPS for 24 h. The culture medium was used for the measurement of IL-6 production using mouse IL-6 ELISA kit (R&D Systems, Inc., Minneapolis, MN, USA).

### 4.9. Statistical Analysis

NO determination, MTT assay, WB, and ELISA data are presented as the mean ± SD. A value of *p* < 0.05 was considered statistically significant. An unpaired *t*-test was used to determine the difference between experimental groups. Different letters represent significant differences among experimental groups, and means with the same letter are not significantly different (*p* > 0.05).

## 5. Conclusions

We searched metabolites of marine organisms for compounds that could inhibit LPS-induced NO production by RAW264.7 cells. PGA_2_ was identified from the MeOH extract of the soft coral *Lobophytum* sp. and was demonstrated to inhibit LPS-induced NO production and suppress the expression of iNOS in LPS-stimulated RAW264.7 cells. LPS-induced degradation and resynthesis of IκBα were not inhibited by short preincubation with PGA_2_, but LPS-induced IκBα degradation was reduced in cells subjected to prolonged treatment with PGA_2_. Furthermore, PGA_2_-inhibited NO production was negated by the addition of an EP4 antagonist, L161982, in LPS-stimulated RAW264.7 cells. These results suggest that PGA_2_ suppresses LPS signaling not by direct inhibition but by the mediation of its receptor and related protein synthesis in RAW264.7 cells. Thus, PGA_2_ was identified as an inhibitor of LPS-induced inflammatory signaling. Further research may reveal the detailed mechanism of PGA_2_ as an anti-inflammatory agent.

## Figures and Tables

**Figure 1 marinedrugs-20-00316-f001:**
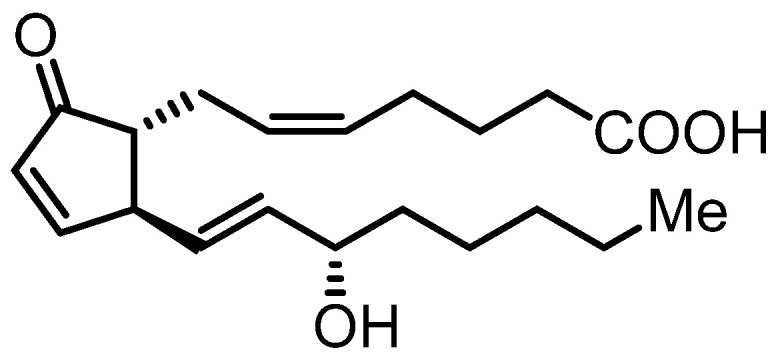
Structure of (15*S*)-prostaglandin A_2_.

**Figure 2 marinedrugs-20-00316-f002:**
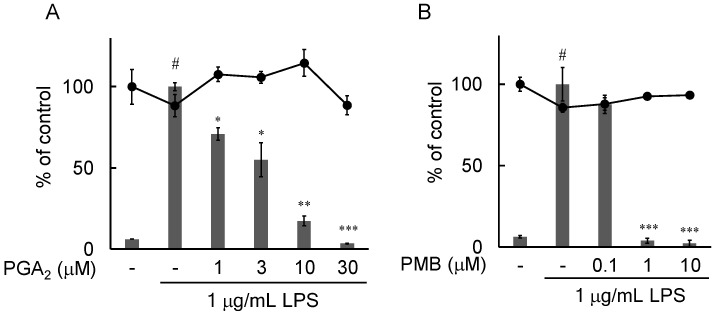
Effects of PGA_2_ on LPS-induced NO production in RAW264.7 cells. Cells were pretreated with the indicated concentrations of PGA_2_ (**A**) or PMB (**B**) for 20 min, followed by treatment with 1 μg/mL LPS for 24 h. NO production in the culture medium was determined using Griess reagent. Columns: NO levels determined by the Griess method; circles: cell numbers determined by MTT assays. Values are the mean ± SD of triplicate determinations. Differences between groups were analyzed using an unpaired *t*-test. ^#^ *p* < 0.005 vs. the control group; * *p* < 0.05, ** *p* < 0.01, *** *p* < 0.005 vs. the LPS-treated group.

**Figure 3 marinedrugs-20-00316-f003:**
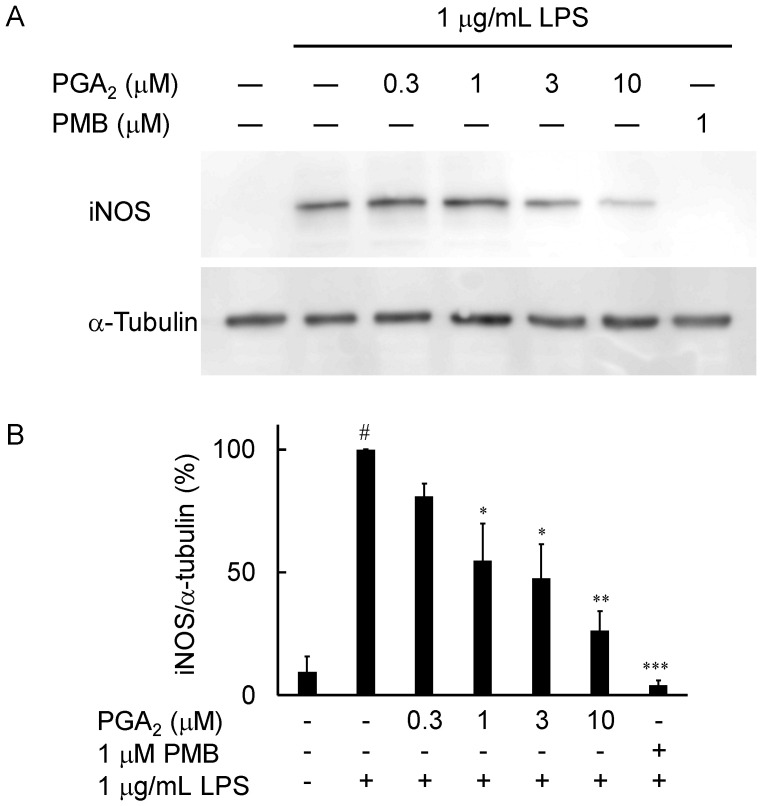
Effects of PGA_2_ on LPS-induced iNOS expression in RAW264.7 cells. (**A**) RAW264.7 cells (1 × 10^6^) were pretreated with the indicated concentrations of PGA_2_ or 1 μM PMB at 20 min prior to exposure to 1 μg/mL LPS for 24 h; then, cell lysates were prepared. Total cellular proteins were resolved by SDS-PAGE, transferred to PVDF membranes, and detected using antibodies specific against iNOS and α-tubulin. (**B**) The density ratios of iNOS to α-tubulin are displayed in a histogram. Values are the mean ± SD of triplicate determinations. Differences between groups were analyzed using an unpaired *t*-test. ^#^ *p* < 0.005 vs. the control group; * *p* < 0.05, ** *p* < 0.005, *** *p* < 0.001 vs. the LPS-treated group.

**Figure 4 marinedrugs-20-00316-f004:**
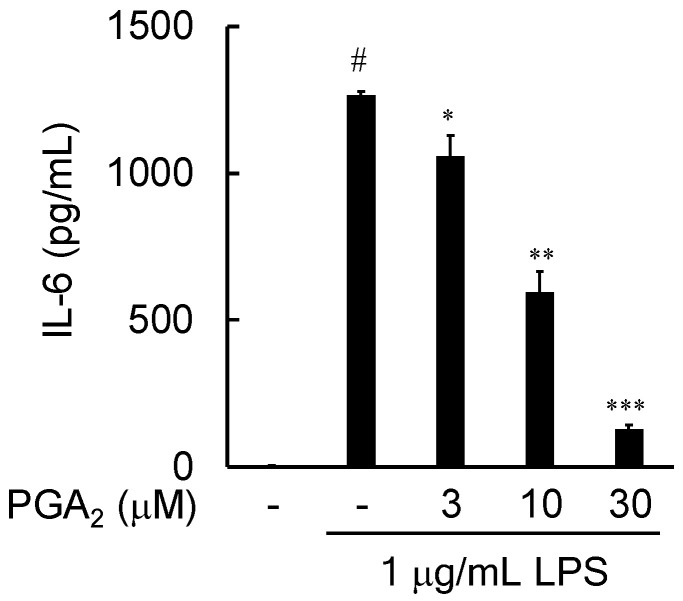
Effects of PGA_2_ on LPS-induced expression of IL-6 in RAW264.7 cells. Cells were pretreated with the indicated concentrations of PGA_2_ for 20 min, followed by treatment with LPS for 24 h. The IL-6 contents in the culture medium were determined by ELISA. Values are the mean ± SD of triplicate determinations. Differences between groups were analyzed using an unpaired *t*-test. ^#^ *p* < 0.0001 vs. the control group; * *p* < 0.05, ** *p* < 0.005, *** *p* < 0.0001 vs. the LPS-treated group.

**Figure 5 marinedrugs-20-00316-f005:**
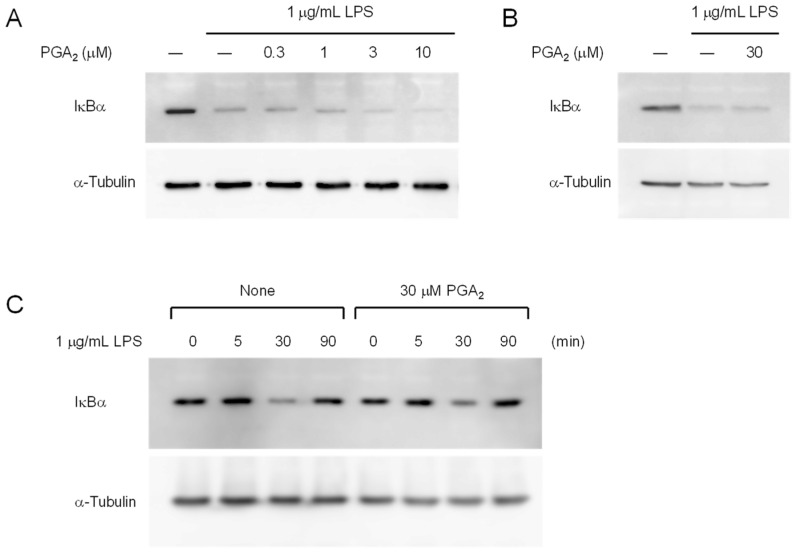
Effects of PGA_2_ on LPS-induced degradation of IκBα. (**A**) RAW264.7 cells (1 × 10^6^) were pretreated with the indicated concentrations of PGA_2_ at 20 min prior to exposure to 1 μg/mL LPS for 30 min, and the cell lysates were analyzed by Western blotting with antibodies against IκBα and α-tubulin. (**B**) RAW264.7 cells (1 × 10^6^) were pretreated with 30 μM PGA_2_ at 20 min prior to exposure to 1 μg/mL LPS for 30 min. Then, the cell lysates were analyzed by Western blotting with antibodies against IκBα and α-tubulin. (**C**) RAW264.7 cells (1 × 10^6^) were preincubated or not with 30 μM PGA_2_ for 20 min, then treated with 1 μg/mL LPS for the indicated periods. The cell lysates were analyzed by Western blotting with antibodies against IκBα and α-tubulin.

**Figure 6 marinedrugs-20-00316-f006:**
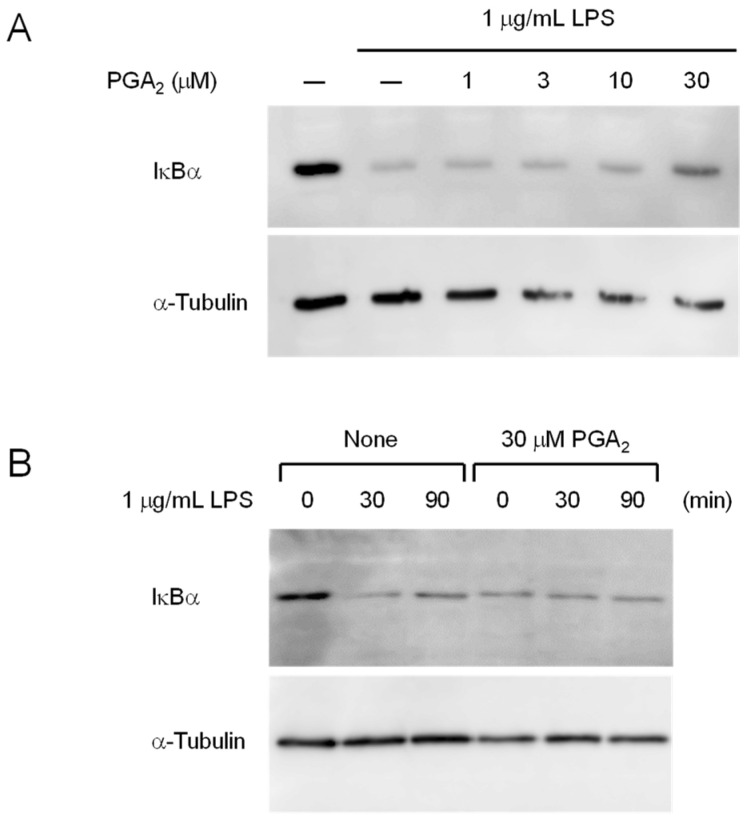
Inhibition of LPS-induced degradation of IκBα by prolonged treatment with PGA_2_. (**A**) RAW264.7 cells (1 × 10^6^) were pretreated with the indicated concentrations of PGA_2_ for 18 h prior to exposure to 1 μg/mL LPS for 30 min; then, the cell lysates were analyzed by Western blotting with antibodies against IκBα and α-tubulin. (**B**) RAW264.7 cells (1 × 10^6^) were preincubated or not with 30 μM PGA_2_ for 18 h, then treated with 1 μg/mL LPS for the indicated periods. The cell lysates were analyzed by Western blotting with antibodies against IκBα and α-tubulin.

**Figure 7 marinedrugs-20-00316-f007:**
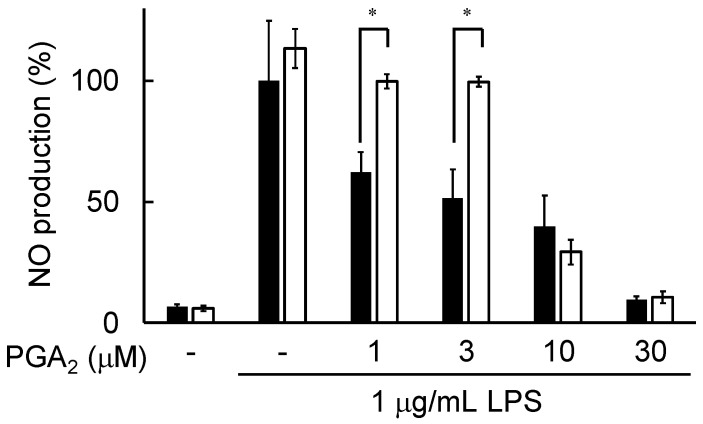
Effects of L161982 on PGA_2_-inhibited NO production in LPS-stimulated RAW264.7 cells. Cells were pretreated (open columns) or not (solid columns) with 10 μM L161982 for 20 min, followed by treatment with the indicated concentrations of PGA_2_ for 20 min. Then, cells were treated with or without 1 μg/mL LPS for 24 h. NO production in the culture medium was determined using Griess reagent. Values are the mean ± SD of triplicate determinations. Differences between groups were analyzed using an unpaired *t*-test. * *p* < 0.05.

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
