# Peer review of "Inhibition of Lipopolysaccharide-Induced Inflammatory Signaling by Soft Coral-Derived Prostaglandin A2 in RAW264.7 Cells"

_marinedrugs, 2022, doi:10.3390/md20050316_

Round 1

Reviewer 1 Report

The manuscript is well-written, however, the author did not clearly show any chromatogram of how the active compound was isolated and purified from the MeOH extract of the soft coral Lobophytum sp. and was identified as  PGA2 since PGA2 is naturally formed by the dehydration of PGE2. In nature, there are four main bioactive prostaglandins PGE2, PGI2, PGD2, and PGF2α.

 PGA2 mostly used in the anti-inflammation test seemed to be the commercial compound not from the soft coral. Why was PGA2 purified from the coral not used or compared with in the same tests? Since It was shown to be an inhibitor of LPS-induced inflammatory signaling in RAW264.7 cells, it may have the same mechanism for the inhibition.

Author Response

>The manuscript is well-written, however, the author did not clearly show any chromatogram of how the active compound was isolated and purified from the MeOH extract of the soft coral Lobophytum sp. and was identified as PGA2 since PGA2 is naturally formed by the dehydration of PGE2. In nature, there are four main bioactive prostaglandins PGE2, PGI2, PGD2, and PGF2α.

Answer: Thank you very much for the warm comment. The description of the purification method of PGA2 in the experimental section regarding was revised. In addition, as pointed out by reviewer 2, the process of proof that the purified compound was identical to PGA2 is shown in the attached file. Furthermore, we have added spectral data of the isolated PGA2 to the supporting information.

>PGA2 mostly used in the anti-inflammation test seemed to be the commercial compound not from the soft coral. Why was PGA2 purified from the coral not used or compared with in the same tests? Since It was shown to be an inhibitor of LPS-induced inflammatory signaling in RAW264.7 cells, it may have the same mechanism for the inhibition.

Answer: Thank you very much for the valuable comment. We have certainly compared the NO production inhibitory activity of PGA2 isolated from the soft coral and purchased PGA2, and obtained the same results as shown in the attached file. Since the amount of PGA2 obtained from the soft coral was very small, this paper shows the results using the purchased products. In addition, based on the comment of reviewer 2, we have confirmed that the NMR spectrum of the isolated PGA2 and that of the purchased PGA2 are the same as shown in the attached file.

Reviewer 2 Report

  1. The Introduction part is not clear, and there is little related introduction to the progress of inhibiting lipopolysaccharide-induced inflammatory signals, especially other marine metabolites. In addition, it is not clear why Soft Coral Lobophytum sp is chosen as the research object.
  2. Many key figure of PGA2 are not provided, such as NMR, HRMS, etc.
  3. It is inappropriate to put 4.3.1. in the Materials and Methods section.
  4. The literature cited in the introduction section is too old.
  5. The 1H NMR data provided at present do not match the structural formula, and the accuracy of the compound deserves further consideration.

Author Response

>1. The Introduction part is not clear, and there is little related introduction to the progress of inhibiting lipopolysaccharide-induced inflammatory signals, especially other marine metabolites. In addition, it is not clear why Soft Coral Lobophytum sp is chosen as the research object.

Thank you very much for the valuable comments. Based on the comments, we have revised the introduction and included a citation of a recent paper on LPS signaling inhibitors derived from marine organisms (Ref. 11 and 12).

The reason for using soft coral as the research object is that the extract of this soft coral was hit as a result of random screening of extracts of multiple marine organisms. This is mentioned in L49 and L188 in the present manuscript.

>2. Many key figure of PGA2 are not provided, such as NMR, HRMS, etc.

Answer: We thank the reviewer’s valuable comment. We added the NMR spectra of the isolated PGA2 in the Supporting Information.

>3. It is inappropriate to put 4.3.1. in the Materials and Methods section.

Answer: Thank you very much for the comment. We moved this information to the Supporting Information.

>4. The literature cited in the introduction section is too old.

Answer: Thank you very much for the valuable comment. Based on the comments, we have changed Ref 4 to a recent review paper. In addition, as we mentioned in the reply to the first comment, I have quoted the recent research results.

>5. The 1H NMR data provided at present do not match the structural formula, and the accuracy of the compound deserves further consideration.

Answer: Thank you very much for the valuable comment. The details of the structure determination are as follows. The isolated compound was speculated to be PGA2 by the NMR analyses. As shown in the table in the attached file, the NMR data of the isolated compound appeared to be consistent with those of PGA2, but not with those of 15-epi-PGA2. However, the reported NMR data of PGA2 was old and only low resolution data was available, so we compared the NMR spectra of the isolated compound and the purchased PGA2 with the specific rotation. As a result, as shown in the attached file, the NMR data and the specific rotation of the isolated compound were in good agreement with purchased PGA2, so the isolated compound was decided to be PGA2. Since the reported spectrum data of PGA2 is only very old, we think it would be informative to include the NMR data in SI of this article.

Reviewer 3 Report

General comment

The manuscript by Ohno et al. reports on the anti-inflammatory properties displayed by PGA2 in LPS-stimulated RAW264.7 cells. The experimental work has been carried out carefully, the results are of interest and the manuscript is easy to follow and generally readable.  I would recommend its acceptance after minor revision, according to the following comments.

Specific comments

Page 1, lines 26-28: Please rephrase

Page 2, line 64, and throughout the manuscript: Please change “dose” with “concentration” as this is not an in vivo study.

Author Response

>General comment

>The manuscript by Ohno et al. reports on the anti-inflammatory properties displayed by PGA2 in LPS-stimulated RAW264.7 cells. The experimental work has been carried out carefully, the results are of interest and the manuscript is easy to follow and generally readable.  I would recommend its acceptance after minor revision, according to the following comments.

Answer: Thank you very much for the kind words.

>Specific comments

>Page 1, lines 26-28: Please rephrase

As the reviewer pointed out, we changed the expression on lines 26-28.

>Page 2, line 64, and throughout the manuscript: Please change “dose” with “concentration” as this is not an in vivo study.

Answer: As the reviewer commented, we changed “dose” to “concentration”.

Round 2

Reviewer 2 Report

I think the manuscript has met the requirements of publication.